# The Potential Therapeutic Effects of Botulinum Neurotoxins on Neoplastic Cells: A Comprehensive Review of In Vitro and In Vivo Studies

**DOI:** 10.3390/toxins16080355

**Published:** 2024-08-13

**Authors:** Delaram Safarpour, Fattaneh A. Tavassoli, Bahman Jabbari

**Affiliations:** 1Department of Neurology, Oregon Health and Science University, Portland, OR 97239, USA; safarpou@ohsu.edu; 2Department of Pathology, School of Medicine, Yale University, New Haven, CT 06520, USA; fattaneh.tavassoli@yale.edu; 3Department of Neurology, School of Medicine, Yale University, New Haven, CT 06520, USA

**Keywords:** botulinum toxin, botulinm neurotoxin, neoplasm, cancer, cancer cells in vitro, in vivo

## Abstract

A systematic review of the literature found fifteen articles on the effect of a botulinum toxin on neoplastic cell lines and eight articles on in vivo neoplasms. The reported in vitro effects rely on high doses or the mechanical disruption of cell membranes to introduce the botulinum neurotoxin into the cell cytoplasm. The potency of the botulinum neurotoxin to intoxicate non-neuronal cells (even cell lines expressing an appropriate protein receptor) is several orders of magnitude lower compared to that to intoxicate the primary neurons. The data suggest that the botulinum toxin disrupts the progression of cancer cells, with some studies reporting apoptotic effects. A majority of the data in the in vivo studies also showed similar results. No safety issues were disclosed in the in vivo studies. Limited studies have suggested similar anti-neoplastic potential for the clostridium difficile. New modes of delivery have been tested to enhance the in vivo delivery of the botulinum toxin to neoplastic cells. Careful controlled studies are necessary to demonstrate the efficacy and safety of this mode of anti-neoplastic treatment in humans.

## 1. Introduction

Botulinum neurotoxins (BoNTs) are widely used for the treatment of a variety of medical conditions. They are considered the drug of first choice for the treatment of cervical dystonia, blepharospasm, and hemifacial spasm [1]. As an inhibitor of acetylcholine release at the neuromuscular junction resulting in decreased muscle tone, BoNTs are potentially powerful agents for the treatment of disabling spasticity in several medical conditions [2]. Furthermore, via the suppression of several major pain transmitters (substance P, calcitonin gene related peptide, and glutamate) [3], the local injection of these agents has been reported to alleviate recalcitrant local pain in a variety of pain syndromes [4,5,6,7]. 

Recent observations indicate a potential role for botulinum toxin therapy in the management of several cancer-related disorders [8]; these include cancer resection/radiotherapy-induced local pain [9,10], the prevention and treatment of post-esophagostomy esophageal stricture [11,12], post-esophagostomy gastroparesis [13,14], post-parotidectomy and post-parotid radiation disorders [15,16], and end-of-life cancer-related disabling symptoms [17].

Over the past 25 years, in vitro and in vivo studies have described the effect of BoNTs on tumor cells, both benign and malignant. Some of these data have been described as part of a descriptive review, including both clinical and laboratory investigations [18]. The purpose of this systematic review is to provide up-to-date information on the potential role of BoNT therapy in the management of benign and malignant tumor cells from the information provided by in vitro and in vivo studies. 

## 2. Study Design 

In this investigation, two independent reviewers (B.J. and F.A.T.) performed a literature search for relevant articles published up to 1 May 2024. The data were cross-verified by a third reviewer (D.S.) to ensure accuracy. The search engines included Med Line, Cochrane library, and Scopus. The search strategy utilized the following terms: botulinum toxin and cancer cells, botulinum toxin and cancer cell line, botulinum toxin and tumor cells, and botulinum toxin and malignant neoplasm. The study selection process started with abstract/title screening followed by full-text review. The search included randomized controlled trials, observational studies, and case-controlled studies. Exclusion criteria consisted of articles in non-English language without an English abstract or an English abstract not providing sufficient information, review articles, letters to the editor, and single case reports. After completion of the search, the total number of articles retrieved and the numbers in each category were presented in a Prisma flow diagram. 

## 3. Results 

This search identified three-hundred-twenty manuscripts. After all the exclusions, twenty-two relevant manuscripts were chosen for the final analysis. Of these, thirteen pertained to “in vitro” and eight described “in vivo” studies (Figure 1) (Table 1 and Table 2). 

## 4. In Vitro Studies [Table 1]

These studies are presented in chronological order of publication. In a study published in 1991 [19], the investigators found that the exposure of PC-12 cells of pheochromocytoma to onabotulinumtoxinA (onaA) and botulinum toxins E and B resulted in a significant reduction in the catecholamine secretion from these cells, mainly due to the function of the light chain of these toxins. Boyd et al. [20] looked at the effect of BoNT-A and BoNT-B upon the beta cells (HIT-15 and RINm5) of the pancreatic tumor insulinoma. Both toxins cleaved over 90% of the SNAP25 and Synaptobrevin present on these cells. The exposure of these cells to BoNT-A and BoNT-B resulted in 90% and 60% drops in the insulin secretion from the tumor cells, respectively. Huang et al. [21] found that the transfection of BoNT-A into the SNAP-25 of HIT-15 pancreatic insulinoma cells can regulate the secretion of insulin from these cells. The authors concluded that BoNT-A has a therapeutic potential for endocrine tumors. Purkiss et al. [22] showed that the presence of SNAP25, VAMP, and syntaxin-1 in undifferentiated human SH-SY5Y neuroblastoma cells and the exposure of these cells to BoNTs A, B, and C significantly changes the secretion of noradrenalin from these cells. Karsenty et al. [23] assessed the effect of onabotulinumtoxinA (onaA) upon prostate (PC-3) and LNCaP cancer cell lines. Both cell lines possess the SV2 receptor (synaptic vesicle glycoprotein2), which is a target of onaA. The exposure to onaA caused significant inhibition of LNCaP proliferation and increased apoptosis of these cells, while there was no significant effect on the PC-3 cells. Compared to the controls, one unit of onaA markedly lowered the level of PSA (prostate-specific antigen) over 28 days. Proietti et al. also confirmed the presence of SV2 receptors on both of the above-mentioned prostate cancer cell lines [24]. Experimenting with a different botulinum toxin-A (IncoA), these authors found a reduction in cell growth after the exposure to that toxin for both cell lines within 96 h of exposure (25% for LNCaP and 20% for PC-3 cells). 

Bandala et al., in two studies [25,26], assessed the effect of onaA on breast cancer cell lines. In the first study [25], they found that onaA exerted more toxicity and apoptosis on cancer cells (T47D) compared to normal cells (MCF10A). In the second study [28], they observed that exposure to onaA reduced the SV2 receptor expression in three different breast cancer cell lines (T47D, MDA-MB-231, and MDA-MB—453). The authors stated that, since onaA can regulate the SV2 expression on breast cancer cells, it could offer a therapeutic advantage when it is used in conjunction with transtuzumab for breast cancer treatment. In contrast to the above studies, however, Cheng et al. [27] reported that 10 units of onaA had no effect on the survival of LNCaP and PC-3 prostate cancer cells. Another group of investigators studied the effect of Herceptin alone, onaA alone, and the combination of onaA and Herceptin on two breast cancer cell lines [38]. The bioconjugation of Herceptin and onaA significantly improved the efficacy of the treatment on both breast cancer cell lines, suggesting a potential advantage of using an immunotoxin treatment for HER-2 positive breast cancers. Rust et al. [29] studied the effect of botulinum toxin C on the two neuroblastoma cell lines—SiMa and SH-SY5Y. Botulinum toxin C that cleaves SNAP25 and Syntaxin1 caused the apoptosis of differentiated neuroblastoma cells. 

Shebl et al. [30] found that both onaA and captopril, when added to the culture media in a dose-dependent manner, inhibited the growth and migration of colon (HCT116) and prostate cancer (DU145) cell lines but did not affect normal cell lines. They also noted that the exposure to onaA raised the tumor protein 53 (p53) level 2.5 times. The authors concluded that the exposure to onaA promotes these tumor cells’ self-destruction, possibly through stimulating p53 gene apoptotic activity. Apkinar et al. [31] exposed neuroblastoma and glioblastoma tumor cells lines (HC-T116 and DU145) to 5 units of OnaA. After 24 h, the cancer cells demonstrated apoptosis and died. The authors attributed this effect to the toxin’s stimulation of the TRPM-2 pathway, resulting in an excess production of mitochondrial oxygen reactive species (ROS). Most recently, Demir et al. [32] examined the potential anti-cancer action of an adjuvant treatment. They found that a combined treatment using onaA and oxaliplatin rendered a synergistic effect, leading to oxidative stress and apoptosis of the colon cancer cells. 

## 5. In Vivo Studies

The eight studies in this category are presented in their chronological order of publication: 

Ansiauox et al. [33] injected onaA (29 units/Kg) into the two tumor models (fibrosarcoma and hepatocarcinoma) in mice’s thighs (at two sites) following the implantation of the tumor cells in the same locations. The oxygenation and perfusion of the growing tumor cells were measured by resonance oximetry and magnetic resonance imaging. The authors found that, following the onaA injection, both the oxygenation and perfusion significantly increased—through the opening of the vascular bed—leading to a better tumor response to radiotherapy and chemotherapy (cyclophosphamide, 50 mg/Kg). Vezdervanis et al. [34] reported a patient with metastatic prostate cancer resistant to polypharmacy (alfuzosin, lanreotide, and dexamethasone). The patient received 1000 units of abobotulinumtoxinA (aboA) to the affected prostate area. After the botulinum toxin injection, the patient was treated with Capecitabine and Finasteride. An ultrasound repeat examination, a month after the toxin injection, disclosed a 30% reduction in the tumor size. In mice with LNCaP and PC3 cancer cells injected in the prostate, Cheng et al. [27], however, noted no effect on the tumor cell growth or prostate size after injecting 2 units of onaA two weeks after the cancer cell inoculation. In another experiment [35], the researchers injected onaA into the greater curvature of the stomach in mice with spontaneous gastric malignant tumors. At 6 months, there was slower development and progression of the tumors as well as fewer inflammatory changes in the injected areas (anterior part of the greater curvature). 

Ulloa et al. [36] studied the effect of BoNT-C1 injection in the glioblastoma. The authors injected U373 glioblastoma cells pretreated with BoNT-C1 into the basal ganglia (striatum) of immunocompromised mice. They found reductions in the glioblastoma growth following the above-mentioned combined injection, which they attributed to the inhibition of the function of syntaxin-1 by the BoNT-C1. 

Another group of investigators [37] studied the pancreatic malignant tumor progression in mice after the injection of MIA-PaCa2 pancreatic cancer cells alone or the cancer cells pretreated with onaA. The group that received cancer cells and onaA together demonstrated significant apoptotic changes in the cancer cells. Based on this observation, the authors concluded that the progression of pancreatic cancers is influenced by their neural environment, and adjuvant therapy with onaA offers a novel therapeutic approach for the management of these malignant tumors. In the above two studies, pretreatment with a botulinum neurotoxin seemingly affected the cells’ ability to implant and establish a tumor. 

Coarfa et al. [38] conducted in vivo experiments with onaA on mice as well as on patients with prostatic cancer. In the mouse experiment, the injection of onaA (10 units) into the prostate before the implantation of prostatic cancer cells (VCaP) resulted in a reduction in the tumor incidence and decreased tumor size. In the human experiment, the authors conducted a clinical trial in which onaA (100 units) was used as a neoadjuvant therapy in those patients affected by prostate cancer before prostatectomy. Similarly, the authors found that, in the human experiment, the injection of onaA into the malignant tumor promoted the apoptosis of the cancer cells. They emphasized the importance of an intact neural environment for the survival of cancer cells. 

Kwak et al. [39] compared the effect of antiPD-1 treatment alone and a combined treatment with onaA (15 u/Kg) upon colon, breast, and lung carcinoma as well as melanoma in mice. Injections were performed in the tumor (onaA) and intraperitoneally (antiPD1). The authors found that, in the B16-F10 (melanoma) syngeneic mouse tumor model, an anti-PD-1 + BoNT/A1 combination treatment lowered the proportion of MDSCs, negated the increased proportion of T_reg_ cells, and elicited a higher number of tumor-infiltrating CD4^+^ and CD8^+^ T lymphocytes into the tumor microenvironment compared to the anti-PD-1 treatment alone. The authors concluded that, in the mouse model of colon carcinoma and melanoma, the combined treatment with anti-PD-1 and onaA has a synergistic anti-tumor effect; there was no beneficial effect on the mouse models of breast and lung cancer, however. 

## 6. Discussion

Botulinum neurotoxins (BoNT) are a group of clostridial toxins with a remarkable molecular structure that allows them, through distinct steps, to reach and enter the cytosol of targeted cells. So far, eight serotypes of BoNT—A, B, C, D, E, F, G, and X—have been identified based on their serological properties. The A and B serotypes are currently in clinical use. Much has been learned during the past 20 years about the details of the molecular structure of these toxins as well as how they reach the nerve cells after intramuscular or subcutaneous injections. The molecular structure of botulinum neurotoxins consists of a heavy chain (100 KD) and a light chain (50 KD) connected by a disulfide bond. The heavy chain of the toxin (HC) has C and N terminals. The C terminal of the heavy chain is responsible for attaching the toxin molecule to the cell surface receptors. For the type-A toxin, the known receptors are polysialoglycoside and synaptic vesicle glycoprotein 2 (SV2); for the type-B toxin, the known receptor is synaptotagmin I/II [40]. The toxin then internalizes into a recycling synaptic vesicle/endosome-like compartment and then enters into an acidified compartment. Finally, only the light chain sub-unit of the botulinum toxin molecule becomes translocated into the cytosol (via the function of the N terminal of HC). The light chain, a zinc endopeptidase, then attaches to a specific SNARE protein inside the cytosol and deactivates it [41]. The SNARE proteins help the vesicles to fuse to the cell membrane, rupture, and release the chemical transmitter [42]. The SNARE protein for BoNT-A is SNAP 25 and for BoNT-B is synaptobrevin (VAMP) [43]. 

The data from the in vitro studies suggest that adding BoNTs (mostly onA) to different cancer cell lines causes apoptosis and reduces the cell growth (Table 1). Most in vivo studies (seven out of eight) slowed the progression of the malignant tumors, improved the response to chemotherapy, and increased the apoptosis of the cancer cells (Table 2). The outlier among these studies is that of Cheng et al. [27] (Table 1 and Table 2), which found no effect on the prostate cancer cells in either in vitro or in vivo studies. The reason for this discrepancy is not clear. The length of exposure and applied dose of BoNT in this study may partly explain the negative results. In their in vitro group (27), the duration of exposure of the cancer cells to BoNT-A was 6 h, which is shorter than in other studies (Table 1). In their in vivo study, the investigators used 2 units of onaA, a dose that is much smaller than what was used in most of the other experiments (10–100 units—Table 2). 

A deleterious effect on cancer cells has also been reported with other genres of clostridia. Nam et al. [44] showed that, in breast and colon cancers, PLC-γl-transformed cells are highly sensitive to the effect of Clostridium Difficile A (CD-A). Observing the apoptosis of these cells (and not of normal cells) after the exposure to CD-A suggested the potential utility of CD-A for treating these two types of cancer. In another study, Prepens et al. [45] showed that adding Clostridium Difficile B to the cell culture inhibits the Fc epsilon-RI-mediated activation of rat basophilic leukemia cells. Other investigators have reported that, in the cell culture, the exposure to C2 and C3 botulinum toxins slowed down the growth and invasiveness of lymphoma T-cells and pancreatic PAN-C1 cells, respectively [46,47]. 

As the core of the malignant tumors is hypoxic and often necrotic and Clostridium Novye (CN) proliferates in hypoxic media, this genre of clostridium attracted the attention of researchers for battling malignancy. Diaz et al. [48] reported CN-NT (non-toxic) spores as potent anti-malignant tumor agents in mice and rabbits. The injection of large doses of CN-NT (non-toxic) to mice destroyed the malignant tumors without causing toxicity. Similar experiences have been reported by others and in different mice cancers [49,50]. Taking advantage of CRISPR technology, the investigators advanced this form of oncological treatment by inserting a tumor-targeting peptide (TTP) into the coat of the CN-NT spores. This further assisted in tumor localization, with better results [51]. 

Botulinum toxins may also influence the growth of cancer cells through affecting the secretion of certain transmitters. By secreting glutamate, the glial cells in the brain control the excitability of their neighboring neurons [52]. It has been shown that, in malignant gliomas, the level of increased secretion of glutamate by normal glial cells can promote the growth of cancer cells [53]. In one investigation [54], the investigators showed that exposure to BoNT-A reduces the production of glutamate by normal glial cells. 

Considering the above in vitro and in vivo data that suggest a potential for BoNT therapy for cancer treatment, the challenge today is to determine how to deliver BoNTs safely into cancer cells without damaging the normal host cells. Over the past few years, attempts to reach this goal have led to a few promising discoveries: 

Fonfria et al. [55] substituted the neuronal binding domain of BoNT-A by an epidermal growth factor ligand (EGF). This novel approach resulted in the selective and successful delivery of bioengineered BoNT-A into the SiMa neuroblastoma cells. 

In both in vitro and in vivo studies, Whitt et al. [56] targeted the SV2 receptors on pancreatic and medullary thyroid cancer cells by a conjugate of the recombinant heavy chain receptor binding domain of BoNT-A (rHCR) and an anti-mitotic agent monomethyl auristatin E (MMAE). This recombinant of BoNT-A selectively targets the SV2 receptors, which, as stated before, are abundantly present on the surface of endocrine cells. In their in vitro study, they noted significant growth suppression of pancreatic and medullary cancer cells after the exposure to the above-mentioned conjugate. Their in vivo study in mice also produced promising results. The treatment with the conjugate of BoNT (rHCR) and MMAE reduced the tumor volume in the mice without producing any adverse effect. 

Botulinum neurotoxins have also been shown to interrupt the pathologic migration and proliferation of non-cancerous cells, for instance keratinocytes in psoriasis [57]. They are currently used as an off-label drug for the treatment of plaque psoriasis [58] 

The application of botulinum toxin therapy for malignant tumors in humans still requires extensive investigations. Although botulinum neurotoxin therapy is generally considered to be safe when performed according to the established guidelines [59,60,61], much remains to be explored and discovered regarding the safety and applicability of BoNT therapy in human malignancy. Future studies are needed and should apply novel technologies (i.e., CRISPR) to explore the potential inclusion of botulinum toxins in the management of human cancers.

## Figures and Tables

**Figure 1 toxins-16-00355-f001:**
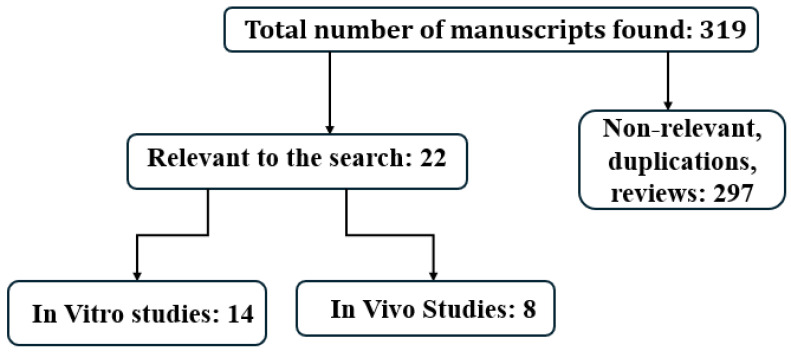
Prisma.

**Table 1 toxins-16-00355-t001:** Botulinum toxin’s effects on different neoplastic cell lines (in vitro studies).

Author and Year	Origin/Type of Tumor	Cell Lines	Type of Toxin	Result
Lomnette et al., 1991 [19]	Pheochromocytoma	PC-12	OnaA	Reduced the secretion of catecholamines
Boyd et al., 1995 [20]	Pancreas	Beta cell line: HIT-15 and RIN-m5F	Ona-A Rima-B	OnaA: >90% inhibition of insulin secretion RimaB: 60% inhibition of insulin secretion
Huang et al., 1998 [21]	Endocrine	Insulin secreting HIT-T15	Ona-A	Regulated insulin secretion
Purkiss et al., 2001 [22]	Neuroblastoma	Shsy-5y	OnaA	Very sensitive
Karsenty et al., 2009 [23]	Prostate	PC3 and LNCaP	Ona-A	Reduced proliferation and increased apoptosis of LNCaP cells
Proietti et al., 2012 [24]	Prostate	PC3 and LNCaP	Inco-A	Reduction in cell growth PC3 25%, LNCaP 20%
Bandala et al., 2013 [25]	Breast	T47D	Ona-A	Greater cytotoxicity to T47D cells than normal cells
Bandala et al., 2015 [26]	Breast	T47D, MDA-MB-231, MDA-MB-453	Ona-A	onaA diminished Sv2 expression in all three cell lines
Cheng et al., 2013 [27]	Prostate	PC3and LNCaP	Ona-A	No effect during day 1 to 6 exposure
Hajighasemlou et al., 2015 [28]	Breast	SK-BR3 and BT-474	Ona-A	Herceptin–onaA conjugate improved Herceptin efficacy against breast cancer cells
Rust et al., 2016 [29]	Neuroblastoma	SiMa and SH-SY-5Y	BTX-C	Caused apoptotic changes only in neuroblastoma cells
Shebl et al., 2019 [30]	Colon and Prostate	HC-T116 and DU145	Ona-A	Inhibitory effect on cell proliferation and ability of cancer cells to migrate
Akpinar et al., 2020 [31]	Neuroblastoma and glioblastoma	DBTRG and SH-SY5Y cells	Ona-A	Apoptosis and death of cancer cells via increase in ROS production
Demir et al., 2024 [32]	Colon	H229	OnaA	Caused oxidative distress and apoptosis of tumor cells

CD: clostridium difficile; ROS: reactive oxygen species.

**Table 2 toxins-16-00355-t002:** Botulinum toxin’s effects on neoplastic cells and neoplasms (in vivo studies).

Author and Year	Location of Cancer	Tested Subject	Type of Toxin and Dose	Results
Ansiauox et al., 2006 [33]	Fiborsarcoma, hepatocarcinoma	Injected into Mouse tumor	onaA, 29 units/Kg	Increased tumor oxygenation and response to chemotherapy
Vezdrevanis et al., 2011 [34]	Prostate	Injected into Patient’s tumor	abo-A, 1000 units	30% reduction in tumor size in 4 weeks.
Cheng et al., 2013 [27]	Prostate	Injected LNCaP and PC3 cancer cells into the prostate of the mice	onaA, 2 units (two weeks later)	Pre-treatment with onaA had no effect on cancer
Zhao et al., 2014 [35]	Stomach	Injected into the greater curvature of the stomach of INS-GNS mice with spontaneous gastric cancer	onaA, 0.1 unit/mouse every month for 6 months.	Slowed development and progression of the cancer
Ulloa et al., 2015 [36]	Brain glioblastoma multiforme (GBM)	Mice, U373 cells with BoNT-C1 injected into striatum	BoNT—C1, 375 pg	Impaired GBM cell proliferation by blocking the function of syntaxin-1
He et al., 2016 [37]	Pancreas cancer	Injected into tumor with MIA-PaCa2 cells treated with OnaA	onaA, 20 units/Kg	Increased apoptosis, decreased malignant tumor size
Coarfa et al., 2018 [38]	Mice Prostate Prostate (Human Clinical trial)	injected into prostate of mouse before implanting VCaP cancer cells Into prostate malignant prostate tumor	onaA, 10 units onaA, 100 units	Reduction in tumor incidence and malignant tumor size Increased apoptosis of cancer cells
Kwak et al., 2023 [39]	Melanoma, colon, lung, and breast carcinoma	Single injection into implanted tumor cells in B16-F10 and MC38 tumor-bearing mice	onaA, 15 units/Kg	Improved anti-cancer activity of PD1 check point blockade

## Data Availability

All data generated and analyzed during this study are included in the published article; further inquiries can be directed to the corresponding author.

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
