# Peer review of "The Potential Therapeutic Effects of Botulinum Neurotoxins on Neoplastic Cells: A Comprehensive Review of In Vitro and In Vivo Studies"

_toxins, 2024, doi:10.3390/toxins16080355_

Round 1
Reviewer 1 Report
Comments and Suggestions for Authors
This review is based on a literature survey on the in vitro and in vivo effects of botulinum toxins on cancer cells. 23 articles were analyzed. The review is well written and as far as I can judge, covers the area well. It complements a review recently published by Toxins, likely by the same authors or group of authors, on the treatment by botulinum toxins for cancer-related disorders (doi: 10.3390/toxins15120689).
Author Response
Thank you for your positive response. We appreciate the time that you have taken to review our manuscript.
Reviewer 2 Report
Comments and Suggestions for Authors
The authors have conducted a systematic literature review of in vitro and in vivo studies of the effects of botulinum neurotoxin on cancer cells. Twenty three articles met the search criteria, 15 reporting effects in cell culture and 8 reporting effects in vivo. The authors report that the majority of these reviewed papers reported that botulinum toxin disrupts progression of cancer cell growth.
Major comments
1. Abstract, line 7: Please add that the reported in vitro effects rely on high doses or mechanical disruption of cell membranes to introduce the botulinum neurotoxin into the cell cytoplasm. The potency of botulinum neurotoxin to intoxicate non-neuronal cells (even cell lines expressing an appropriate protein receptor) is several orders of magnitude lower compared to primary neurons.
2. Abstract, lines 7-8: Typo “The data suggests that botulinum toxin-A disrupts…” should read “The data suggest that botulinum toxin disrupts…”; several of the reviewed papers describe effects of other botulinum neurotoxin serotypes, not just type-A.
3. Abstract, line 8: Please change “…and causes apoptosis” in the sentence “…disrupts progression of cancer cells and causes apoptosis” to “…with some studies reporting apoptotic effects”. It seems a little too strong to have an unqualified statement that the data suggest botulinum toxin causes apoptosis since no mechanism is suggested. Evidence and mechanistic rationale is strongest for botulinum neurotoxin serotype-C1 which acts on an additional intracellular target compared to other serotypes.
4. Abstract, lines 10-11: Please change the sentence “New modes of delivery have been developed to enhance in vivo delivery…” to “New modes of delivery have been tested to enhance in vivo delivery…”, because, as yet, none have been clinically successful.
5. Table 2, line 56: Typo, table is incorrectly labelled as in vitro studies, should be in vivo.
6. In vitro studies, line 63: Typo “…over 90% of SNAP25 and Syntaxin1…” should read “…over 90% of SNAP25 and synaptobrevin…” The botulinum nerurotoxin serotypes used in this study do not act on Syntaxin1.
7. In vitro studies, lines 65-67: “In another study [21]…Botulinum toxin C2…” I recommend excluding this paper [Verschueren and others, 1995, Eur J Cell Biol], because Botulinum toxin C2 is a member of a different toxin family from the other botulinum neurotoxins in this review.
8. In vitro studies, line 70: Typo “…presence of SNAP25, WAMP, and Syntaxin-1…” should read “…presence of SNAP25, VAMP, and Syntaxin-1…”
9. In vitro studies, lines 72-74: “In another study [24], the investigators found that botulinum toxin C3, via inhibiting Rho…” I recommend excluding this paper [Kusama and others, 2001, Cancer Res], because Botulinum toxin C3 is a member of a different toxin family from the other botulinum neurotoxins in this review.
10. In vitro studies, line 99: Please change “…a potential advantage of using such a combination treatment…” to “…a potential advantage of using an immunotoxin treatment…”. This is to make clear that the paper describes an immunotoxin, in which botulinum neurotoxin was chemically crosslinked to the Herceptin antibody and not a mixture of the two individual agents.
11. In vivo studies line 138: “…injected U373 glioblastoma cells pretreated with BoNT-C1 into the basal ganglia…” Please make it clear that pretreatment of cells with botulinum neurotoxin (before injection into the in vivo model) is very different from injection into an established tumor. The pre-treatment may have affected the cell’s ability to implant and establish a tumor.
12. In vivo studies lines 142-144: Same comment as above (pre-treatment of cells prior to injection is different from injection into an implanted tumor).
13. In vivo studies line 160: Please make it clear that in [Kwak and others, 2023, Immunol Invest], the botulinum neurotoxin was only injected into the tumor. The antiPD1 antibody was injected intraperitoneally.
14. Discussion, lines 180-181: “…toxin attaches itself to the SV2 receptor which then opens like a channel and allows the whole toxin molecule to enter the cytosol.” This is incorrect. SV2 does not open a channel and the whole toxin molecule does not enter the cytoplasm. Botulinum neurotoxins bind to two cell surface receptors (in most cases a protein and a polysialoganglioside receptor), then become internalised into a recycling synaptic vesicle/endosome-like compartment, then trafficked into an acidifying compartment, then just the light chain sub-unit of the botulinum neurotoxin molecule becomes translocated into the cytosol. Please correct the statement and provide citations so that readers can check for themselves the literature describing the molecular mechanism by which botulinum neurotoxins intoxicate cells.
15. Discussion, lines 181-183: “Previous studies suggest that SV2 channels are present on different cancer cells [21,22,28] especially on malignant endocrine tumor cells.” This statement is incorrect. There is no evidence that SV2 forms channels. Please correct.
16. Discussion, lines 184-185: “The data from the in vitro studies suggest that adding BoNTs (mostly onA) to different cancer cell lines causes apoptosis and reduces cell growth (Table1)”. This reviewer disagrees with this statement, I don’t believe that is what the data suggest. There are reports of apoptosis but it isn’t clear if it is a direct effect of the neurotoxin or secondary to the methods used to introduce it into the cytoplasm of a cell type that it doesn’t easily intoxicate. There is no mechanistic rationale for how botulinum neurotoxin-A might act to activate apoptotic signalling. There is some rationale for how botulinum neurotoxin-C1 might do this, since it cleaves an additional intracellular substrate (Syntaxin-1) in addition to SNAP25 and, indeed, two of the papers listed in table 1 tested Botulinum neurotoxin-C1.
Minor comments
1. Introduction, line 30: Typo “Some of these data has been…” should read “Some of these data have been…”
2. Study design, line 37: Typo “The data was cross verified…” should read “The data were cross verified…”
3. Study design, line 45: Typo “…sufficient information., review…” should read “…sufficient information, review…”
4. Study design, line 47: typo “...presented in a Prisma.” should read “…presented in a Prisma flow diagram.”
5. In vitro studies, lines 110-111: Typo “The authors contributed this effect…” should read “The authors attributed this effect…”
Author Response
Reviewer 2-
Thank you for very useful and constructive comments. In therevised, marked copy, changed are in red
Major comments-
- Abstract, line 7: Please add that the reported in vitro effects rely on high doses or mechanical disruption of cell membranes to introduce the botulinum neurotoxin into the cell cytoplasm. The potency of botulinum neurotoxin to intoxicate non-neuronal cells (even cell lines expressing an appropriate protein receptor) is several orders of magnitude lower compared to primary neurons.
Answer: your points are covered in Lines 7-11 of the revised abstract
- Abstract, lines 7-8: Typo “The data suggests that botulinum toxin-A disrupts…” should read “The data suggest that botulinum toxin disrupts…”; several of the reviewed papers describe effects of other botulinum neurotoxin serotypes, not just type-A.
Answer: corrected in the Line 11 of the revised abstract
- Abstract, line 8: Please change “…and causes apoptosis” in the sentence “…disrupts progression of cancer cells and causes apoptosis” to “…with some studies reporting apoptotic effects”. It seems a little too strong to have an unqualified statement that the data suggest botulinum toxin causes apoptosis since no mechanism is suggested. Evidence and mechanistic rationale is strongest for botulinum neurotoxin serotype-C1 which acts on an additional intracellular target compared to other serotypes.
Answer: Changed according to your suggestion in Line 12 of the abstract
- Abstract, lines 10-11: Please change the sentence “New modes of delivery have been developed to enhance in vivo delivery…” to “New modes of delivery have been tested to enhance in vivo delivery…”, because, as yet, none have been clinically successful.
Answer: changed in line 15 of the abstract
- Table 2, line 56: Typo, table is incorrectly labelled as in vitro studies, should be in vivo.
Answer - Corrected
- In vitro studies, line 63: Typo “…over 90% of SNAP25 and Syntaxin1…” should read “…over 90% of SNAP25 and synaptobrevin…” The botulinum nerurotoxin serotypes used in this study do not act on Syntaxin1.
Answer: Thank you for correcting our error. The error is corrected (page 4, line 72)
- In vitro studies, lines 65-67: “In another study [21]…Botulinum toxin C2…” I recommend excluding this paper [Verschueren and others, 1995, Eur J Cell Biol], because Botulinum toxin C2 is a member of a different toxin family from the other botulinum neurotoxins in this review.
Answer: Following your advice excluded from the table. Discussed briefly in discussion (Page7, line 211).
- In vitro studies, line 70: Typo “…presence of SNAP25, WAMP, and Syntaxin-1…” should read “…presence of SNAP25, VAMP, and Syntaxin-1…”
Answer: corrected in page 5 line 77
- In vitro studies, lines 72-74: “In another study [24], the investigators found that botulinum toxin C3, via inhibiting Rho…”I recommend excluding this paper [Kusama and others, 2001, Cancer Res], because Botulinum toxin C3 is a member of a different toxin family from the other botulinum neurotoxins in this review.
Answer: Aso excluded from the table. Discussed briefly in discussion ( Page 7, line 212).
- In vitro studies, line 99: Please change “…a potential advantage of using such a combination treatment…” to “…a potential advantage of using an immunotoxin treatment…”. This is to make clear that the paper describes an immunotoxin, in which botulinum neurotoxin was chemically crosslinked to the Herceptin antibody and not a mixture of the two individual agents.
Answer: changed on page 5, line 102
- In vivo studies line 138: “…injected U373 glioblastoma cells pretreated with BoNT-C1 into the basal ganglia…” Please make it clear that pretreatment of cells with botulinum neurotoxin (before injection into the in vivo model) is very different from injection into an established tumor. The pre-treatment may have affected the cell’s ability to implant and establish a tumor.
Answer - A statement was added to the manuscript regarding your point (page 6, lines 151 and 152.
- In vivo studies lines 142-144: Same comment as above (pre-treatment of cells prior to injection is different from injection into an implanted tumor).
Answer – covered by the statement on page 6, lines 151-152
- In vivo studies line 160: Please make it clear that in [Kwak and others, 2023, Immunol Invest], the botulinum neurotoxin was only injected into the tumor. The antiPD1 antibody was injected intraperitoneally.
Answer: clarified (page 6, line 164)
- Discussion, lines 180-181: “…toxin attaches itself to the SV2 receptor which then opens like a channel and allows the whole toxin molecule to enter the cytosol.” This is incorrect. SV2 does not open a channel and the whole toxin molecule does not enter the cytoplasm. Botulinum neurotoxins bind to two cell surface receptors (in most cases a protein and apolysialoganglioside receptor), then become internalised into a recycling synaptic vesicle/endosome-like compartment, then trafficked into an acidifying compartment, then just the light chain sub-unit of the botulinum neurotoxin molecule becomes translocated into the cytosol. Please correct the statement and provide citations so that readers can check for themselves the literature describing the molecular mechanism by which botulinum neurotoxins intoxicate cells.
Answer: At the suggestion of another reviewer, we added a paragraph describing molecular structure of botulinum toxin. In That using some of your suggested wordings, we have described the internalization and function of botulinum neurotoxins. Two new references were provided (pages lines 180-192). . The erroneous statement about SV2 was omitted
- Discussion, lines 181-183: “Previous studies suggest that SV2 channelsare present on different cancer cells [21,22,28] especially on malignant endocrine tumor cells.” This statement is incorrect. There is no evidence that SV2 forms channels. Please correct.
Answer: The erroneous sentenc was omitted. SV2 receptor used instead of SV2 channel (lines 183-184)
- Discussion, lines 184-185: “The data from the in vitro studies suggest that adding BoNTs (mostly onA) to different cancer cell lines causes apoptosis and reduces cell growth (Table1)”. This reviewer disagrees with this statement, I don’t believe that is what the data suggest. There are reports of apoptosis but it isn’t clear if it is a direct effect of the neurotoxin or secondary to the methods used to introduce it into the cytoplasm of a cell type that it doesn’t easily intoxicate. There is no mechanistic rationale for how botulinum neurotoxin-A might act to activate apoptotic signalling. There is some rationale for how botulinum neurotoxin-C1 might do this, since it cleaves an additional intracellular substrate (Syntaxin-1) in addition to SNAP25 and, indeed, two of the papers listed in table 1 tested Botulinum neurotoxin-C1.
Answer: We respect your opinion as an expert in the field. We are merely saying what the authors of those four manuscripts in table 1 suggested and described.
Minor comments
- Introduction, line 30: Typo “Some of these data has been…” should read “Some of these data have been…” –
Answer : corrected Line 34
- Study design, line 37: Typo “The data was cross verified…” should read “The data were cross verified…”
Answer: corrected Line 41
- Study design, line 45: Typo “…sufficient information., review…” should read “…sufficient information, review…”
Answer : Corrected Line 49
- Study design, line 47: typo “...presented in a Prisma.” should read “…presented in a Prisma flow diagram.”
Answer: Corrected: Lines 51,52
- In vitro studies, Typo “The authors contributed this effect…” should read “The authors attributed this effect…”
Answer: Corrected. Page 5, Line 114 of the revised manuscript.
We thank you again for your remarkable suggestions- we learned a lot from you

Reviewer 3 Report
Comments and Suggestions for Authors
In this systematic review of literature, the authors expose the in vitro and in vivo studies of the effects of clostridial toxins, basically botulinum toxins, on a series of tumor and cancer cells, considering the possibility to be used in therapy for cancer treatments. The general idea is well exposed although some additional information would be useful to fully understand the review.
For instance, authors could include bibliography (significant reviews) as reference for the molecular mechanism of action of botulinum neurotoxins. Alternatively, it should be convenient to include a very brief comment on the general structure of botulinum neurotoxins, its domains and the functions associated to each one, mainly the domains related to binding and the specific effect on nerve terminals, and the lytic (proteolytic) effect on the secretory molecular machinery, through its specific proteolytic activity on SNARE proteins (the inhibitory effect on the secretory molecular machinery was initially identified in nerve terminals, and lately basically in all exocytosis dependent release models, providing the lytic domain of the neurotoxins gain access to the secretory machinery (see comments below on the effect of botulinum neurotoxins on secretory celll ines).
This would explain some of the results included in the review, concerning the effect of botulinum neurotoxins on secretory cell lines, and its possible specificity in those cells bearing botulinum neurotoxins receptors.
Additional comments:
Table 1.- The year of reference Cheng et al., should be 2013, as in table 2.
Line 58-59.- Reference 19 is related to inhibition of norepinephrine release on PC12 in permeabilized, no intact, cells, mainly by botulinum toxin serotypes E and B (in addition to A), and due to the light chain of the toxins. It would be convenient to add this information in the text.
Line 63.- Reference 20, SNAP25 and synaptobrevin were cleaved by BoNT-A and BoNT-B respectively, but not syntaxin as it is stated in the text (line 63). Cells were electroporated for the toxins to gain access to intracellular targets.
Line 66 and line 73.- Neither botulinum toxin C2 nor botulinum toxin C3 are neurotoxins. These toxins have different targets and mechanism of action than botulinum neurotoxins, which are the subject of the manuscript. Both toxins (botulinum toxins C2 and C3) are ADP-ribosylating toxins affecting the cytoskeleton. This comment might also include the “in vivo” approach mentioned in the discussion, lines 202-210 related to the necrotizing effect of Clostridium Novyi. Accordingly, the title should be changed (toxins instead of neurotoxins), or these part of the text can be removed, or an explanation why clostridial toxins but no neurotoxins should be provided.
Author Response
Reviewer 3:
Thank you for your constructive comments- In the revised marked copy, change are in red
Authors could include bibliography (significant reviews) as reference for the molecular mechanism of action of botulinum neurotoxins. Alternatively, it should be convenient to include a very brief comment on the general structure of botulinum neurotoxins, its domains and the functions associated to each one, mainly the domains related to binding and the specific effect on nerve terminals, and the lytic (proteolytic) effect on the secretory molecular machinery, through its specific proteolytic activity on SNARE proteins (the inhibitory effect on the secretory molecular machinery was initially identified in nerve terminals, and lately basically in all exocytosis dependent release models, providing the lytic domain of the neurotoxins gain access to the secretory machinery (see comments below on the effect of botulinum neurotoxins on secretory cell lines).
Answer: A paragraph was added to the text to explain the cell attachment, internalization and action of the light chain of the toxin once it enters the cytosol (Pages lines 180 to 192). Per your advice, two references were provided in the text explaining structure of neurotoxins and their effects on the secretory machinery of the cells (references 42 and 43 in the revised manuscript).
Additional comments:
Table 1.- The year of reference Cheng et al., should be 2013, as in table 2.
Answer: The year was corrected in the table.
Line 58-59.- Reference 19 is related to inhibition of norepinephrine release on PC12 in permeabilized, no intact, cells, mainly by botulinum toxin serotypes E and B (in addition to A), and due to the light chain of the toxins. It would be convenient to add this information in the text.
Answer: per your advice, this information was added to the manuscript (page 4, lines 68 and 69 of the revised manuscript)
Line 63.- Reference 20, SNAP25 and synaptobrevin were cleaved by BoNT-A and BoNT-B respectively, but not syntaxin as it is stated in the text (line 63). Cells were electroporated for the toxins to gain access to intracellular targets.
Answer: Thank you for mentioning this error. It is corrected in line 72 (page 4) of the revised manuscript
Line 66 and line 73.- Neither botulinum toxin C2 nor botulinum toxin C3 are neurotoxins. These toxins have different targets and mechanism of action than botulinum neurotoxins, which are the subject of the manuscript. Both toxins (botulinum toxins C2 and C3) are ADP-ribosylating toxins affecting the cytoskeleton. This comment might also include the “in vivo” approach mentioned in the discussion, lines 202-210 related to the necrotizing effect of Clostridium Novyi. Accordingly, the title should be changed (toxins instead of neurotoxins), or these part of the text can be removed, or an explanation why clostridial toxins but no neurotoxins should be provided.
Answer: Per your recommendation and advice of one other reviewer we removed data on C2 and C3 botulinum toxins from the tables. We briefly discussed data from these two toxins and other two non-neurotoxins (difficile and, Novye) in discussion regarding their effects on cancer cells and tumors (Page 7, lines 210 and 212).

Reviewer 4 Report
Comments and Suggestions for Authors
Please consider including the potential role of botulinum toxin in treating psoriasis, given the similarities between psoriasis and neoplastic cells, such as hyperproliferation and altered differentiation. BoNT's inhibition of neurogenic inflammation by impeding the release of substance P and CGRP has led to its 'off-label' use in inflammatory skin disorders like psoriasis. This addition could broaden the understanding of BoNT's therapeutic potential across different pathological conditions.
Please see this paper: Popescu MN, Beiu C, Iliescu MG, Mihai MM, Popa LG, Stănescu AMA, Berteanu M. Botulinum Toxin Use for Modulating Neuroimmune Cutaneous Activity in Psoriasis. Medicina (Kaunas). 2022 Jun 16;58(6):813. doi: 10.3390/medicina58060813. PMID: 35744076; PMCID: PMC9228985.
Comments on the Quality of English Languagethere are several areas where the phrasing could be more concise and the flow improved for better readability. Some grammatical issues include incorrect verb tense usage and inconsistent terminology, such as referring to botulinum toxin-A as both "BoNT-A" and "onaA." Additionally, punctuation errors are present, like the unnecessary comma in "The search engines included, Med Line, Cochrane library and Scopus." Furthermore, certain sentences could benefit from rephrasing for clarity, such as "Their mechanism of action as inhibitor of acetylcholine release at the neuromuscular junction leading to decreased muscle tone makes them a powerful treatment agent," which could be revised to enhance readability. Addressing these issues will enhance the overall clarity and professionalism of the manuscript.
Author Response
Reviewer 4
Thank you for your constructive comments. In the revised marked copy, changes are in red
Please consider including the potential role of botulinum toxin in treating psoriasis, given the similarities between psoriasis and neoplastic cells, such as hyperproliferation and altered differentiation. BoNT's inhibition of neurogenic inflammation by impeding the release of substance P and CGRP has led to its 'off-label' use in inflammatory skin disorders like psoriasis. This addition could broaden the understanding of BoNT's therapeutic potential across different pathological conditions.
Please see this paper: Popescu MN, Beiu C, Iliescu MG, Mihai MM, Popa LG, Stănescu AMA, Berteanu M. Botulinum Toxin Use for Modulating Neuroimmune Cutaneous Activity in Psoriasis. Medicina (Kaunas). 2022 Jun 16;58(6):813. doi: 10.3390/medicina58060813. PMID: 35744076; PMCID: PMC9228985.
Answer: Thank you for your constructive suggestions. Following your suggestion, we have added a short paragraph in discussion about the use of botulinum neurotoxins in psoriasis [page 8, lines 244-246 – references 57 and 58]
Comments on the Quality of English Language
there are several areas where the phrasing could be more concise and the flow improved for better readability. Some grammatical issues include incorrect verb tense usage and inconsistent terminology, such as referring to botulinum toxin-A as both "BoNT-A" and "onaA." Additionally, punctuation errors are present, like the unnecessary comma in "The search engines included, Med Line, Cochrane library and Scopus." Furthermore, certain sentences could benefit from rephrasing for clarity, such as "Their mechanism of action as inhibitor of acetylcholine release at the neuromuscular junction leading to decreased muscle tone makes them a powerful treatment agent," which could be revised to enhance readability. Addressing these issues will enhance the overall clarity and professionalism of the manuscript.
Answer: Following your advice, we improved the grammar and clarified long sentences. For example, page 1, lines 21 to 23.

Round 2
Reviewer 2 Report
Comments and Suggestions for Authors
Many thanks to the authors for addressing all the points raised.
One further suggestion is to make a small addition to Discussion, line 185: and add the word “protein” to the sentence “…type B toxin the known receptor is synaptotagmin I/II”, such that the sentence reads “…type B toxin the known protein receptor is synaptotagmin I/II”; this is because the type B neurotoxin also binds to a polysialoglycoside receptor, in addition to synaptotagmin I/II.